# Botulinum Neurotoxins as Two-Faced Janus Proteins

**DOI:** 10.3390/biomedicines13020411

**Published:** 2025-02-08

**Authors:** Silvia Chimienti, Maria Di Spirito, Filippo Molinari, Orr Rozov, Florigio Lista, Raffaele D’Amelio, Simonetta Salemi, Silvia Fillo

**Affiliations:** 1Istituto di Scienze Biomediche Della Difesa, 00184 Rome, Italy; silvia.chimienti@persociv.difesa.it (S.C.); maria.dispirito@persociv.difesa.it (M.D.S.); filippo.molinari@esercito.difesa.it (F.M.); florigio.lista@esercito.difesa.it (F.L.); 2Dipartimento di Scienze Cliniche e Medicina Traslazionale, Università di Roma Tor Vergata, 00144 Rome, Italy; 3Dipartimento di Sanità Pubblica e Malattie Infettive, Sapienza Università di Roma, 00185 Rome, Italy; 4Food and Agriculture Organization, 00153 Rome, Italy; orr.rozov@gmail.com; 5Independent Researcher, 00162 Rome, Italy; raffaele.damelio@gmail.com; 6Azienda Ospedaliero-Universitaria S. Andrea, 00189 Rome, Italy

**Keywords:** botulinum neurotoxins, *Clostridia*, foodborne botulism, infant botulism, wound botulism, iatrogenic botulism, biosecurity, BIG-IV, HBAT, ebselen

## Abstract

Botulinum neurotoxins are synthetized by anaerobic, spore-forming bacteria that inhibit acetylcholine release at the level of the neuromuscular and autonomic cholinergic junctions, thus inducing a series of symptoms, the most relevant of which is flaccid paralysis. At least seven serotypes and over 40 subtypes are known, and they are among the most poisonous natural substances. There are different forms of botulism according to the route of contamination, but the clinical manifestation of descending symmetric flaccid paralysis is consistent, regardless of the route of contamination. It is very severe and potentially lethal. The induced paralysis lasts as long as the toxin is active, with variable length, according to the serotype of the toxin. This transient activity, as well as the precise mechanism of action, are the basis for the rationale behind use of the toxin in therapy for several clinical conditions, particularly, spastic conditions, as well as chronic migraine and axillary hyperhidrosis. The toxin has also been approved for the reduction in facial wrinkles; all these clinical applications, coupled with the toxin’s risks, have earned botulinum the title of a two-faced Janus protein. No approved vaccines are currently available, andthe only approved antidotes are the human specific intravenous immunoglobulins for infant botulism and the heptavalent equine immunoglobulins/(F(ab’)^2^ for adults. Nanobodies, which show great promise, may penetrate neuronal cells to inactivate the toxin within the cytoplasm, and Ebselen, a non-toxic, economic, small-molecule inhibitor, has the characteristic of inhibiting the toxin irrespective of the serotype.

## 1. Introduction

Botulinum (from the Latin word “botulus” meaning sausage, a food first identified as the source of botulism) neurotoxins (BoNTs) are proteins synthetized by anaerobic, rod-shaped, spore-forming bacteria of the genus *Clostridium* (from the Greek word “Kloster”, which means spindle shaped) [1], such as *Clostridium botulinum* (classified as *C. botulinum* Groups I–III) and *C. botulinum* Group IV also known as *C. argentinense*. *C*. *botulinum* Groups I and II are associated with human botulism, whereas *C. botulinum* Group III is associated with botulism in various animal species; Group IV strains form BoNT/G, a serotype which has not been definitively associated with human or animal botulism [2,3]. Some strains of *C. baratii* and *C. butyricum* synthetize BoNT/F7 and E4 and E5, respectively, all associated with human botulism [3].

Seven immunologically different BoNT serotypes are known, named from A to G, and over 40 subtypes [4] (Table 1). The BoNT serotypes have been discovered over the course of half a century, with BoNT/A and BoNT/B first identified in 1919 [5] and BoNT/G in 1970 [6]. Recent progress in genomic sequencing has revealed multiple subtypes (designated with Arabic numbers), which can be recognized by the same antiserum but exhibit sequence variations greater than 2.6% [7]. In 2016, for the first time, a BoNT-like metalloprotease was identified in *Weissella oryzae*, a non-clostridial species [8], and such observation has even been made with *Enterococcus faecium* and *Chryseobacterium piperi* [9]. Additionally, several mosaic toxins have been identified. For instance, a “type H” was reported in 2013, but it was later classified as a mosaic toxin, with its light chain sharing ∼80% identity with the light chain of a BoNT/F5 subtype and its heavy chain sharing ∼84% identity with the heavy chain of BoNT/A1. This neurotoxin is variously described as BoNT/FA, BoNT/H, and BoNT/HA [10]. Moreover, the genomic sequencing of *C. botulinum* strain 111 revealed a potentially novel BoNT called BoNT/X (GenBank no: BAQ12790.1), which cleaves SNARE (Soluble NSF Attachment Protein Receptor) family members on a novel single cleavage site on VAMP2 (between Arg66 and Ala67) [11]. *C. botulinum* strain 111 is a dual-toxin-producing strain, with *bont/x* localized on the chromosome and *bont/b2* localized on a plasmid. BoNT/X appears to be either silent or non-toxic to vertebrates, including humans [12].

BoNT is the most powerful natural poison, with an estimated LD_50_ of 0.09–0.15 μg if injected intravenously/intramuscularly in a man of approximately 70 kg, 0.70–0.90 μg if absorbed by inhalation, and 70 μg if ingested [13]. Serotypes A, B, E, and, more rarely, F are pathogenic for humans, who may be infected through different routes of contamination. Not all the seven BoNTs have the same virulence, however, because the time of recovery after intoxication is variable, and it depends on the dose and the type of BoNT with the following order: BoNT/A~BoNT/C > BoNT/B~BoNT/D~BoNT/F~BoNT/G > BoNT/E [14]. Toxin serotypes are categorized based on the toxin’s ability to be recognized and neutralized by specific antibodies. The difference in amino acid level may be up to 70% among the seven serotypes [7] and generally lower, but up to almost 50%, among the subtypes [3]. Strains from different clostridial groups can produce the same toxin serotype (e.g., BoNT/F is produced by Groups I, II, and V), and subtypes of the same serotype can differ in biological activity [7]. BoNTs can cause botulism, a potentially fatal disease, which is mostly present in wild and domesticated animals and, more rarely, may even be observed in humans [15].

## 2. Genetics, Structure, and Mechanism of Action of BoNT

BoNTs are encoded by *bont* genes located on either chromosomal, plasmid, or phage elements. The *bont* gene is always located adjacent to the non-toxic non-haemagglutinin gene (*ntnha*). BoNTs are secreted as progenitor toxin complexes (PTCs) together with non-toxic non-haemagglutinin (NTNHA) without a few neurotoxin-associated proteins (NAPs) (M complex, progenitor of BoNT). This arrangement reduces the exposure of the BoNT to external damaging agents, thus suggesting that NTNHA shields and protects the BoNT molecule from proteolytic and other chemical attacks [2]. The *bont* and *ntnha* genes are near other genes that encode either haemagglutinin or OrfX proteins. These proteins associate with the BoNT–NTNHA heterodimer and are also thought to have a protective role [2]. Structurally, all different toxin serotypes are made of two polypeptide chains covalently linked by a single conserved disulfide bond: a heavy chain (H, 100 kDa) and a light chain (L, 50 kDa). Functionally, the H chain can be divided into two main domains: the C-terminal binding domain (HC domain), which mediates the specific binding of the toxin to the plasma membrane of motor neurons, and the N-terminal translocation domain (NC domain), which forms a permeation channel, allowing the translocation of the L chain into the cytoplasm. So, the L chain, that represents the catalytic domain of the protein, becomes ready to exert its enzymatic function through the interchain disulfide bond reduction [2]. The specific cleavage of the L chain represents the catalytic domain of the protein and contains a zinc endopeptidase activity which targets and cleaves the SNARE proteins in the neuronal cell, involved in the neuro-exocytosis nanomachine [16,17].

BoNT can enter the body via the gastrointestinal system, via inhalation, or from infected wounds. Once absorbed, BoNT is transported via the bloodstream and distributed throughout host tissues. Due to its size, the toxin cannot cross the blood–brain barrier. It can travel via retrograde axonal transport into the central nervous system (CNS) [1], but typically acts on peripheral neurotransmission affecting the neuromuscular junctions and inducing paralysis of the nerve terminals [2]. This is a multi-step process that essentially leads to the blockade of neurotransmitter acetylcholine release from the neuronal terminal through the specific cleavage of SNARE proteins. In more detail, botulinum toxin binds rapidly and with high affinity to the presynaptic plasma membrane of skeletal and autonomic cholinergic nerve terminals. BoNTs have evolved a unique binding mechanism involving sequential interactions with two independent receptors. Initially, all BoNTs bind to poly-sialo-gangliosides (PSGs), leading to the accumulation of the toxins in unmyelinated areas. This facilitates the interaction with a second receptor, which usually is the intraluminal serotype-specific domain of a synaptic vesicle (SV) protein [18]. This binding is responsible for the internalization of the toxin inside the SVs. Following this, the acidification of the vesicle lumen caused by the vesicular ATPase proton pump occurs, creating a pH gradient that drives the accumulation of neurotransmitters via the vesicular neurotransmitter transporter. The acidic pH of the endosome facilitates BoNT and membrane structural change. The HC allows the translocation of the partially or completely unfolded LC from the lumen to the cytosolic side of the vesicle membrane. Once in the cytosol, the LC refolds and dissociates after the reduction of the interchain disulfide bond and specifically cleaves different SNARE protein targets [15,19]. The mechanism of action of BoNTs for inducing paralysis consists of blocking the release of acetylcholine at the neuromuscular synapsis by cleaving molecular components of the protein complex SNARE, composed of VAMP (vesicle-associated membrane protein)/Synaptobrevin, Syntaxin, and SNAP-25 (synaptosomal-associated protein of 25 kilodaltons). SNARE proteins allow the opening of membrane vesicles containing acetylcholine and the fusion with the membrane of the pre-synaptic neuronal cell, thus releasing acetylcholine at the level of the synaptic cleft, inducing muscle activation and contraction [2]. BoNTs B, D, F, and G cleave synaptobrevin, whereas serotypes A, E, and C cleave SNAP-25, with C even cleaving syntaxin [13] (Figure 1).

Cleaving these proteins prevents the formation and/or function of the SNARE complex and, consequently, the neurotransmitter release to the neuromuscular junction, leading to a prolonged functional blockade of the nerve terminal. The BoNTs do not kill the intoxicated neurons, but rather paralyze them in a completely reversible way. The duration of paralysis depends on the type of BoNT, the dose, and the animal species, thus allowing a total recovery of the intoxicated individuals even after several months, provided that respiratory support is administered during the paralytic period [2]. Among the BoNTs, the duration of the BoNT/A1-induced neuroparalysis is the longest (3–4 months for human skeletal terminals and 12–15 months for autonomic cholinergic nerve terminals), while the L chain of BoNT/E1 has the shortest duration (paralysis lasting about 2–4 weeks) [16]. The absolutely powerful and precise functional activity of the BoNTs is on the basis of both the severity of the clinical picture of the intoxication and conversely their possible use as drugs, particularly, for spastic conditions, thus making them “Janus two-faced” molecules.

## 3. Spreading of *Clostridia* in the Animal World

*Clostridia* are present in the soil, and only vertebrate animals are susceptible to BoNT intoxication, while invertebrates, such as insects, maggots, worms, and shellfish, lack PSG receptors, thus resulting in resistance to intoxication. The vegetative form of the *Clostridia* is generally present in the anaerobic environment of decomposing animal cadavers, where they find favorable conditions for growth and BoNT release. On decomposing organic matter, insects may deposit eggs and, together with maggots and other invertebrates, may disseminate the intoxication, when they are eaten by fish and birds in a self-amplifying cycle [15]. This environmental ecology allows us to understand why not only vegetables but also domesticated and wild animals may be intoxicated, thus contributing to the most common forms of botulism: foodborne and infant botulism.

The large diffusion of *Clostridia* in the environment is even associated with their spore-forming nature. Spores from different *Clostridium* species are among the most resistant life forms known. Ubiquitously present in soil and aquatic environments, they can persist for extended periods. These spores exhibit significant resistance to various agents, including extreme temperatures, pressure changes, UV and gamma radiation, enzymatic digestion, and oxidizing agents. Generally, spores are much more resistant to the various agents than the growing cells of the same species. Spores of different strains, species, and genera can exhibit large differences in resistance to various agents. Their structure, made of several layers (such as the exosporium, coat, outer membrane, cortex, germ cell wall, inner membrane, and core), plays a major role in spore resistance. Spores can be killed by damage to several components, including DNA, the spore’s inner membrane, or proteins in the spore core [20]. Chemically, *C. botulinum* spores exhibit high resistance to common disinfectants. High-level disinfectants such as sodium hypochlorite, hydrogen peroxide, and peracetic acid are necessary to achieve sporicidal effects, with efficacy dependent on specific parameters, including concentration, exposure time, and the presence of organic matter, which can significantly diminish sporicidal activity [21,22]. For example, *C. botulinum* spores are inactivated by 0.1% sodium hypochlorite within 5 min [23]. Thermal resistance is another defining feature, with spores surviving standard boiling conditions (100 °C) for extended periods. Effective thermal inactivation typically requires temperatures of 121 °C for at least 3 min, achievable through commercial pressure canning methods. This approach ensures a 12-log reduction in spores, a benchmark standard for food safety [21,24]. Additionally, physical methods such as ionizing radiation have shown efficacy against *C. botulinum* spores. However, the required radiation doses may adversely affect food quality, limiting its application. In meat products, sodium nitrite effectively inhibits spore germination and growth, serving as a critical safety additive in food preservation [21,22]. High-pressure processing (HPP) at 600–900 MPa combined with high temperatures (80–110 °C) can inactivate *C. botulinum* spores. The effectiveness depends on the type of bacteria, growth stage, number of bacteria, and optimal growth temperature. It also depends on the food’s pH, moisture content, and other properties, as well as the treatment conditions [24]. *Clostridia* under hostile environmental conditions, such as low pH, extremes of temperature, oxygen, desiccation, and ultraviolet radiation, transform into spores. The most heat-resistant spores are from the *Clostridium* Group I serotypes and subtypes, followed by the Group III and lastly by the Group II ones [1].

Vegetative bacterial growth and BoNT generation occur in the presence of anaerobiosis, low-salt levels, high water activity, and a pH above 4.6 and in environmental conditions that may be found in deep wounds, in the immature infant intestine, and in the gut of adults with depleted microbiota [1]. BoNTs are inactivated when exposed to 85 °C for at least 5 min [1]. The transition from the spores to vegetative forms occurs when the so-called germinants, represented by small nutrient molecules, interact with germinant receptors located in the inner membrane of the spores. This allows the monovalent cations and dipicolinic acid to be released, and hydrolysis is carried out by lytic enzymes, thus resulting in rehydration and vegetative form outgrowth [1].

## 4. BoNT for Therapeutic or Cosmetic Use

Based on the understanding of their precise, pharmacological mechanism of action, it was reasoned in the 1980s that BoNTs could be useful in the treatment of chronic spastic conditions, such as blepharospasm, dystonias with possible tremors, and hemifacial spasm. Only BoNT/A and BoNT/B have been considered, due to their longer-lasting activity. In 1989, the Food and Drug Administration (FDA) approved BoNT/A for the treatment of strabismus, blepharospasm, and hemifacial spasm [25]. In 1991, a review article reported the optimal results for the approved conditions associated with substantial safety, with generally acceptable, mild, and transient adverse events, and identified additional disorders, which could have benefited from the therapeutic use of BoNT/A [26]. It should be highlightedthat most medical conditions for which BoNT/A is indicated did not have any possibility of effective treatment before their approval [27]. Afterwards, even additional indications, such as the prophylaxis of chronic migraine, for which the rationale for using BoNT/A is less immediately clear than in the spastic forms, probably being associated with the reduction in neurotransmitters release, including the calcitonin gene-related peptide (CGRP), from the peripheral, nociceptive, snf C-fiber nerve endings by BoNT/A, have been approved [28]. Currently, FDA-approved therapeutic indications for BoNT/A are as follows:Treatment of overactive bladder (OAB) with symptoms of urinary incontinence, urgency, and frequency, in adults who have an inadequate response to or are intolerant of an anticholinergic medication;Treatment of urinary incontinence due to detrusor overactivity associated with a neurologic condition, e.g., spinal cord injury (SCI), multiple sclerosis (MS), etc., in adults who have an inadequate response to or are intolerant of an anticholinergic medication;Treatment of neurogenic detrusor overactivity (NDO) in pediatric patients 5 years of age and older who have an inadequate response to or are intolerant of anticholinergic medication;Prophylaxis of headaches in adult patients with chronic migraine (≥ 15 days per month with headache lasting 4 h a day or longer);Treatment of spasticity in patients 2 years of age and older;Treatment of cervical dystonia in adult patients, to reduce the severity of abnormal head position and neck pain;Treatment of severe axillary hyperhidrosis that is inadequately managed by topical agents in adult patients;Treatment of blepharospasm associated with dystonia in patients 12 years of age and older;Treatment of strabismus in patients 12 years of age and older.

The three main BoNT/A therapeutic products (OnabotulinumtoxinA, AbobotulinumtoxinA, and IncobotulinumtoxinA) currently available are not interchangeable due to differences in manufacturing [29]. These differences primarily concern the purification method—crystallization for OnabotulinumtoxinA and chromatography for the other two products—the composition of the final product (OnabotulinumtoxinA and AbobotulinumtoxinA contain inactive complex proteins, with a final molecular weight of ~900 kDa and <500 kDa, respectively, whereas IncobotulinumtoxinA consists solely of the ~150 kDa toxin), the potency of the three products (determined using manufacturer-specific assays), and the dosage units. OnabotulinumtoxinA and IncobotulinumtoxinA are available in 50-, 100-, and 200-unit formulations, while AbobotulinumtoxinA is available in 300- and 500-unit formulations [29]. However, even for products with identical dosage units, they are not interchangeable, as indicated in the drug labels, due to differences in potency—these are biological products rather than chemically identical compounds [29].

In addition to these BoNT/A products, BoNT/B has also been approved for therapeutic use. Specifically, RimabotulinumtoxinB, has been approved only for the treatment of cervical dystonia [30].

The therapeutic indications listed above are not all present in all the three BoNT/A products reported above; moreover, they, as well as DaxibotulinumtoxinA and PrabotulinumtoxinA, have also been approved for aesthetic indications, such as the reduction of facial wrinkles. A Cochrane analysis of these five most common BoNT/A products has found a four-week reduction in wrinkles but noted a probable increased risk of ptosis [31] (Table 2).

The use of BoNTs in therapy has represented a step forward for the treatment of medical conditions for which no therapy was present or surgical treatment which could be avoided in most cases. However, the proliferation of indications, and of off-label uses, particularly in aesthetic medicine, together with the uncontrolled production of unlicensed counterfeit preparations [32], frequently acquired on the internet, without any rigorous control, has allowed a new form of contamination, iatrogenic botulism, to emerge and proliferate. The easy possibility of acquiring large quantities of uncontrolled BoNTs, frequently containing an amount of BoNT higher than what is officially reported on the label, also raises concerns regarding biosecurity [4].

## 5. Forms of Human Botulism—Epidemiology

The WHO identifies different kinds of botulism: foodborne botulism, infant botulism, wound botulism, inhalational botulism, and also adult intestinal botulism and iatrogenic botulism. All types of botulism are potentially fatal and are considered medical emergencies. No person-to-person transmission of botulism has ever been reported [1,33]. However, whatever the route of contamination, the resulting clinical picture is always the same, characterized by an afebrile, descending flaccid paralysis, which may be lethal if not treated, due to the paralysis of the respiratory muscles, leading to respiratory failure and death. The incubation time is variable and dependent on the amount of ingested contaminated food and the type of contamination [4]. Symptoms initially involve cranial nerves, presenting as diplopia, blurred vision, dizziness, dysarthria, dysphagia, dysphonia, and dry or sore mouth and throat, along with fixed mydriasis, without sensory deficits [34], followed by generalized weakness, palpebral ptosis, and dyspnea, because the respiratory muscles are interested. However, even at very advanced clinical stages, the patient remains clear-headed and oriented, because BoNTs may not pass through the blood–brain barrier [4].

### 5.1. Foodborne Botulism

Foodborne botulism, which is the ingestion of pre-formed BoNT, may be due to the consumption of a variety of food, including low-acid preserved vegetables, fish, including canned tuna and salted, fermented and smoked fish, and meat products, including ham and sausages [35]; the most dangerous foods are the home-canned ones and the local traditional foods [34]. Though uncommon, store-bought foods can also be contaminated with botulinum toxin [36]. Groups I and II *C. botulinum* are associated with outbreaks of human foodborne botulism. BoNT/A and BoNT/B are the most prevalent serotypes, with cases of BoNT/E and BoNT/F being rarely reported [37]. Animal outbreaks tend to be associated with BoNT/C and BoNT/D toxin serotypes, although equine botulism is more commonly related to BoNT/B [1]. The incubation time is variable, typically 12–72 h, depending on the amount of contaminated food ingested [1]. Historically, foodborne botulism was the most prevalent; however, preventive measures in food preparation, storage, and distribution have significantly reduced its incidence in recent decades, particularly in the USA [1]. In case of foodborne botulism, the diagnosis may be facilitated by the early presence of gastro-enteric symptoms in approximately 70% of patients [35], which are lacking in the other forms of botulism. The disease is rare, and even when it provokes an outbreak, the diagnosis may be late, and this may compromise survival, thus increasing lethality, which globally ranges from 5% to 10% [33].

### 5.2. Infant Botulism

Infant botulism is caused by the ingestion of spores by infants of 1 week to 12 months of age. Contrary to foodborne botulism, where the pre-formed toxin is ingested in a single episode, here, there is continuous intra-intestinal toxin production due to clostridial colonization of the intestine. This is attributable to immature gut physiology and underdeveloped gut microbiota, which allows *C. botulinum* to flourish and produce the toxin that causes disease [1]. Most cases arise from the consumption of untreated natural food;honey, and corn syrup, for example, are frequently found to contain a high concentration of spores [38]. Typically, the symptoms, characterized by the so-called “floppy babies” with difficulty to swallow or suck, begin 18–36 h after having ingested the contaminated food [39]. Even powdered baby formula and household dust may be contaminated by *C. botulinum* spores [1].

### 5.3. Wound Botulism

Wound botulism is due to contamination of a wound by spores, which may rarely be due to accidental contamination of the wound or, more frequently, is commonly observed in injecting drug users (IDUs). The conditions necessary for toxin production, which are typically found in the wound environment, are strict anaerobiosis, pH > 4.6, and a high protein concentration [40]. Wound botulism was first reported in 1951 [41] and systematically analyzed in 1973, with the description of nine cases, four occurring before 1971 and five between 1971 and 1972, in a period in which wound botulism was rare and only due to the accidental wound contamination by spores [42]. In this study, it was found that the incubation period was higher than that in foodborne botulism, ranging from 4 to 14 days, but the clinical severity and the lethality (4/9 deaths 44%) were similar. The increasing prevalence of IDUs has resulted in a net increase in wound botulism [1], and even an outbreak of nine cases of wound botulism among IDUs was described in California, all treated with heptavalent polyclonal antibodies and with one death (11%) [43].

### 5.4. Adult Intestinal Botulism

Adult intestinal botulism is a rare condition caused by clostridial colonization in subjects who have taken broad-spectrum antibiotics for a long time, or have undergone gastrointestinal surgery, or have inflammatory bowel diseases [1]. Recently, a review describing 24 cases of adult intestinal botulism up to 2020 has been reported [44]. The lethality was 8/24 (33%), 7 due to BoNT/A and 1 to BoNT/F; in all the cases, BoNT/A was present in 12, BoNT/B in 7, BoNT/F in 5, and BoNT/E in 2. Adult intestinal botulism typically presents as isolated cases and is often underdiagnosed.

### 5.5. Iatrogenic Botulism

Iatrogenic botulism is a consequence of the improper administration of excessive doses of BoNTs during therapeutic use. This may happen as a consequence of the use of an unknown BoNT concentration, because the product is counterfeit and not officially acquired, or for the proliferation of off-label uses and poor awareness of the high risk of the used product for health on the part of both the patients and health care workers [45].

### 5.6. Inhalational Botulism

Inhalational botulism is very rare and has only been described as an accidental laboratory exposure or resulting from the deliberate dissemination of aerosolized toxins as biological weapons [13].

### 5.7. BoNT as a Biological Weapon

BoNTs have long been considered ideal biological weapons. This is due to a number ofcharacteristics, including their potent toxicity, severity of induced clinical presentation, and their colorless, tasteless, and odorless nature, making themhardly identifiable; also notable is the relative ease of production, transport, and dissemination, and their potent activity via multiple routes, such as oral, inhalational, and injectional ones with a rather short incubation time ranging from a few hours to a few days, depending on the serotype, dose, and route of contamination [4,13]. Based on these characteristics and the relevant consequences for public health in the case of botulism, BoNTs were categorized by the Centers for Disease Control and Prevention (CDC) in 1999 among category A biological agents, the most dangerous categorization [46]. In 1930’s Manchuria, the head of Japanese Unit 731, which had the task of studying biological weapons, admitted to feeding cultures of Clostridium botulinum to prisoners, to lethal effect [13]. In the same period, other countries were actively engaged in the study of biological weapons, including France, the UK, Canada, and the USA [47]. During the Second World War, the USA produced over one million doses of botulinum toxoid vaccine for immunizing allied soldiers preparing to invade Normandy on D-Day [47]. Although the Biological and Toxin Weapons Convention prohibits offensive research and production of biological weapons, two signatory states, Iraq and the Soviet Union (which was even a co-depositary state), continued producing BoNTs as biological weapons for a long time. BoNT was one of the agents tested at the Soviet site Aralsk-7 on the Vozrozhdeniye Island in the Aral Sea [13]. A former scientist from the civilian Russian bioweapons program reported that the Soviets had tried splicing the BoNT gene from Clostridium botulinum into other bacteria [13]. Moreover, scientists with thisbackground were reportedly recruited, after the collapse of the former Soviet Union, by the US government listed as “state sponsors of terrorism”, such as Iran, Iraq, North Korea, and Syria, which are suspected of developing BoNTs as offensive biological weapons [13]. Regarding Iraq, after the First Gulf War, the United Nations Special Commission on Iraq understood that Iraq had produced 19,000 L of concentrated BoNT, 10,000L of which were already loaded into weapons; this amount would have been capable of killing approximately three times the whole world population by inhalation [13]. In Japan, between 1990 and 1995, the religious sect known as Aum Shinrikyo attempted to spread at least 3-fold aerosolized BoNTs unsuccessfully in Tokyo downtown and at US military installations in Japan [4,13]. The lack of success has been attributed to the defective preparation of the aerosol [4], similar to the lack of success of the same sect when it attempted to spread aerosols of anthrax spores, which were not previously weaponized in particles smaller than 5 μ, thus being unable to enter lung alveoli.

The inhalation route of BoNT contamination is not very efficient, and the BoNT may easily be degraded under sunlight in 1–3 h, with an estimated decay rate of 1–4%, according to weather conditions and the dispersal pattern [48]. Other theoretical meansof BoNT dispersion may be by injection, thus contaminating food, beverages, and medicines, however, this appears to be impractical, considering the rigorous controls in the phases of preparation and distribution. As far as water supply is concerned, treatment with chlorine may inactivate the BoNT and the exposure to air and to sunlight may induce its degradation [4]. One possibility which appears theoretically practicable is the acquisition of BoNT/A for therapeutic use through the internet, where it is possible to find unlicensed counterfeit products containing doses higher than declared [32], but only for use against a single target [4].

### 5.8. Epidemiology

Human botulism is a rare disease, especially in developed countries, where the food and water control are very rigorous [1]. In the five-year period from 2013 to 2017, 547 human cases of botulism and 17 deaths were reported from 22 European Countries, with the highest number of cases (130) being notified by Italy, followed by Romania (84), Poland (74), and France (58). In the same period, in the USA, 900 cases and 15 deaths were notified [4]. In the five-year period of 2018–2022, 422 cases of botulism were reported to the European Centre for Disease Control and Prevention from 17/30 European countries, with an overall notification rate of 0.02/10^5^. Italy still reported the highest number of cases (147), followed by Romania (71), France (46), and Poland (40) [49]. A global review identified 6932 cases reported from 59 world countries in the period January 2000–January 2023 with a global case fatality rate of 1.37%. However, a large underreporting was hypothesized; in fact, based on the comparison with the USA, it was estimated that 88.71% of cases were unreported in 2016 [50]. The high prevalence of foodborne botulism compared with the infant botulism form, the wound form, and the adult intestinal colonization is witnessed by the Italian epidemiology in the period 1986–2015, when 466 confirmed cases of botulism were observed, 90.4% of which were foodborne, 7.7% were infant, 1.3% were due to wound contamination, and 0.6% were a consequence of adult intestinal colonization [51]. However, the epidemiological situation is largely dependent on the country, and in the USA, in 2019, it was possible to observe that, out of 201 total confirmed cases of botulism, 152 (75.6%) were represented by infant botulism, 29 (14.4%) by wound botulism, and only 19 (9.45%) by foodborne, with the last one case occurring due to other causes [52].

Even for iatrogenic botulism, real epidemiological data are lacking due to probable underreporting, considering that, in some countries, the notification of this form of botulism is not mandatory, and the physicians who have administered the drug have no interest in notifying, in order to avoid personal responsibility. Moreover, in some countries, iatrogenic botulism should only be notified to the specialized agencies as an adverse event, thus preventing any reliable data collection [53]. Nevertheless, in the concentrated time period of February–April 2023, a large outbreak of iatrogenic botulism provoked by BoNT serotype A, off-label intragastrically administered in Turkey for the treatment of obesity as a substitute for bariatric surgery in 87 patients coming from Turkey, but even from Germany, Switzerland, France, and Austria, was described [53]. In 2022, the WHO published a medical product alert on five falsified batches of Dysport (BoNT serotype A-haemagglutinin complex) found in five countries—Jordan and Turkey (May), Kuwait and the UK (June), and Poland (July) [54]. Iatrogenic botulism, therefore, may be estimated to occur more frequently than it has been officially reported, probably as a consequence of counterfeit products [55], extensive and growing off-label use, more frequently for cosmetic than medical indications [56], and poor awareness of the high risk of the used product for health on the part of both, the patients and the health care workers.

In Table 3, the different forms of botulism are reported.

## 6. Diagnosis

Diagnosis is not easy, considering that botulism is rare and its clinical presentation may partially mimic other conditions, including stroke, myasthenia gravis, Guillain–Barré syndrome, Lambert–Eaton myasthenic syndrome, and intoxication from different substances [4,13,34]. The clinical characteristics of botulism, which should be underlined in the differential diagnosis, are prominent cranial nerve palsies relative to milder weakness and hypotonia in regions below the neck, the symmetry, and the lack of sensory nerve damage [13]. Moreover, in the case of foodborne botulism, a gastro-enteric symptomatology, with nausea, vomiting, abdominal pain or discomfort, diarrhea, followed by constipation, may precede the onset of neurological symptoms [34]. In wound botulism, fever may be present before the neurologic phase, in contrast with foodborne botulism which is afebrile [42].

Although routine laboratory tests do not usually help in diagnosis, electromyographic analysis does [13]. However, the standard diagnostic test is the mouse bioassay (MBA), using type-specific antitoxin to protect mice from BoNT present in the sample; this assay demonstrates high sensitivity and specificity and may detect as low as 0.03 ng of BoNT, typically within 1–2 days [13]. In parallel, culture in anaerobiosis of the biological samples collected (gastric aspirates, fecal samples, food, etc.) should be carried out, with results typically available within 7–10 days. The culture of gastric aspirates and of fecal samples may be of help even in the case of contamination through the inhalational route [13]. The MBA has long been the gold standard for BoNT detection. Clinical specimens are injected into mice, and their responses are observed over 24–48 h for signs of botulism, such as respiratory distress and paralysis. According to Lebeda et al., this assay can detect all BoNT serotypes (A–G) with an exceptionally low limit of detection (LOD) of approximately 0.1 LD_50_/mL [57]. However, its ethical concerns and limitations, such as prolonged turnaround times and inability to distinguish between active and inactive toxins, have spurred the development of alternative methods. The Endopeptidase Mass Spectrometry (Endopep-MS) utilizes substrate cleavage specific to BoNT serotypes, followed by mass spectrometric analysis. Garland et al. reported its high specificity for serotypes A, B, E, and F, with an LOD as low as 0.05 ng/mL [58]. This assay’s rapidity (4–6 h) and serotype differentiation capabilities make it a valuable tool, although its reliance on specialized equipment and technical expertise limits widespread application. The Real-Time Polymerase Chain Reaction (RT-PCR) detects BoNT gene sequences in clinical and environmental samples. As described by the CDC guidelines, it is highly specific for serotypes A, B, E, and F and achieves an LOD of ~100 copies of DNA per reaction [36]. While providing rapid results (2–4 h), it cannot confirm toxin activity, making it a supplementary rather than a standalone diagnostic tool. The Enzyme-Linked Immunosorbent Assay (ELISA) uses antibodies to detect BoNT proteins. Rossetto et al. highlight its utility in identifying serotypes A, B, E, and F with LODs ranging from 10 to 50 pg/mL [14]. Despite its advantages of speed and non-reliance on animal models, ELISA is susceptible to false positives due to antibody cross-reactivity. The neuromuscular junction (NMJ) models mimic BoNT-induced synaptic blockade at the neuromuscular junction. According to Azarnia Tehran et al., NMJ models demonstrate an LOD of ~0.1 ng/mL for serotypes A and B [59]. They offer an ethical alternative to MBA, though they require further validation for routine diagnostic use.

Recently, a paper-based antibody-free electrochemical sensor able to simultaneously measure BoNT/A/C combined with a smartphone-assisted potentiostat for reliable, easy, and on-site sustainable measure of BoNTs has demonstrated promising results in preliminary testing [60]. Once validated in larger studies, it could serve as an economical, field-deployable test for public health and biosecurity emergencies.

Table 4 summarizes the characteristics of the diagnostic tests reported above.

## 7. Prophylaxis and Therapy

### 7.1. Vaccines

Vaccine prophylaxis in a rare disease as botulism is only indicated for single categories of workers, including exposed laboratory workers and the military, whereas the vaccine is not recommended for the general population for two primary reasons: the low incidence of the disease and the concern that immunization against BoNTs could limit the therapeutic use of BoNT/A or B [61]. Moreover, the existence of different serotypes makes the preparation of a toxoid vaccine more difficult and not comparable to tetanus and diphtheria toxoids, for which a single serotype is known. In addition, the immunization schedule is rather long and laborious (the intramuscular administrations at 0, 2, and 12 weeks, followed by yearly boosters). However, such a toxoid vaccine, developed in the 1920s, was sufficiently protective, but quite reactogenic, and the protective antibody titers tended to wane after a few years, leaving only a small minority of individuals protected. For all these considerations, the pentavalent (A/B/C/D/E) toxoid vaccine, which was prepared in the 1970s and was never FDA-approved, but distributed by the CDC under the investigational new drug (IND) application, in 2011 was stopped [62].

At the start of this century, a monovalent BoNT/F toxoid vaccine was prepared in the UK for US Army Medical Research Institute of Infectious Diseases (USAMRIID) and was tested in volunteers, demonstrating safety and immunogenicity via intramuscular and subcutaneous administration [63]. In 2002, in Japan, a tetravalent A/B/E/F toxoid vaccine was prepared and demonstrated to be safe and effective [64], and the same vaccine has recently been obtained through a new preparation from protein M [65].

More recently, the issue of a preventative anti-botulinum toxin vaccine has been faced again, trying to overcome the problems which have been met with the toxoid vaccine. Additionally, the need to protect laboratory workers at risk of infection/intoxication and of having available adult immunized people as plasma donors for collecting human anti-BoNTs immunoglobulins was considered. The vaccine uses the recombinant HC, which is the C-terminal binding domain of the H chain of the BoNT. Such a vaccine was found to be safe and able to stimulate a 4-fold increase in neutralizing antibodies anti-BoNT/A and B in volunteers, previously immunized with the pentavalent BoNT/A–E toxoid vaccine, who received the recombinant vaccine as a booster. This vaccine can be used to protect laboratory workers exposed to the risk of infection/intoxication and for recruiting immunized plasma donors for obtaining anti-BoNT human immunoglobulins for passive immunization of infants with infant botulism [66]. However, more promising seems to be a candidate vaccine which uses a catalytically inactive recombinant HC holoprotein [67]. In spite of these studies, no approved vaccine against BoNT is currently available.

### 7.2. Passive Immunotherapy

#### 7.2.1. Human Botulism Immune Globulin Intravenous (BIG-IV or BabyBIG)

BIG-IV is an FDA-approved human-derived immunoglobulin, specifically designed to treat infant botulism caused by BoNT serotypes A and B. By neutralizing circulating toxins, it prevents their binding to neuronal receptors. Clinical studies highlight its efficacy in significantly reducing the hospital stay duration (from 5.7 weeks to 2.6 weeks), mean duration of intensive care (by 3.2 weeks), and ventilator dependency (by 2.6 weeks) when administered early, preferably within three days of symptom onset. Its safety profile is robust, making it the preferred treatment for infant cases. BIG-IV reduces hospital stays and ventilation dependency in infants, thus preventing paralysis progression. However, BIG-IV is ineffective once the toxin has entered neurons and is not indicated for serotypes beyond A and B [68].

#### 7.2.2. Heptavalent Botulism Antitoxin (HBAT)

HBAT is an equine-derived antitoxin FDA-approved in 2013 for treating botulism caused by serotypes A through G. Unlike BIG-IV, its broader serotype coverage makes it suitable for various forms of botulism, including wound and foodborne cases in adults. Most commercial products are pepsin-treated, so that the resulting molecule is Fc-deprived F(ab’)^2^ and less reactogenic because it is unable to activate the complement and more capable of entering small spaces, due to its lower molecular weight, but with a shorter half-life and the incapacity to cross the placenta. HBAT works by neutralizing circulating toxins but, similar to BIG-IV, cannot reverse effects in neurons once toxins are internalized. HBAT is produced by Emergent BioSolutions, Winnipeg, Manitoba, Canada, Inc., in vials of 20–50 mL, containing an anti-toxin amount dependent on the serotype as follows: A (4500 IU), B (3300 IU), C (3000 IU), D (600 IU), E (5100 IU), F (3000 IU), and G (600 IU), with the International Units determined by the mouse neutralization assay [34]. Early administration prevents progression to respiratory failure and systemic paralysis. However, its equine origin increases the risk of hypersensitivity reactions, such as serum sickness, necessitating careful patient monitoring [34,69]. Historical data enabled to ascertain an overall rate of adverse events (including hypersensitivity and serum sickness of 9–17% and anaphylaxis of 1.9%) [34]. However, in the previous decades, the recommended dose was two to four times higher than that currently administered. The incidence of hypersensitivity related to administration of a single vial is reduced; the risk of serum sickness may be approximately 1–4%. One vial should be slowly infused, after having mixed at the ratio 1:10 with physiologic solution, possibly within 24 h of symptom onset, to have the best results of reduction in hospital/intensive care stay and mortality [34]. However, the reduction in lethality in botulism, from >60% in the early 20th century to the current <5% [70], occurred more as a consequence of improvement in intensive care assistance than for the timely use of passive immunization, which was already available then [69]. The effect of BoNT is transient, thus mechanical ventilation is necessary up to the end of activity of BoNT.

In addition to HBAT, even trivalent A/B/E equine immunoglobulins are available in Europe, from two different producers, the first from Biomed, Krakow, Poland, in vials of 10 mL with 5000 IU of anti-A and anti-B and 1000 IU of anti-E, the second from Behring, Marburg, Germany, in bottles of 250 mL, with 187,500 IU of anti-A, 125,000 IU of anti-B, and 12,500 IU of anti-E. The dose of trivalent A/B/E equine immunoglobulins that should be administered, it is from 1 to 5 vials of the Biomed’s antitoxin, and 2 bottles of the Behring’s antitoxin [34].

#### 7.2.3. Monoclonal Antibodies

BabyBIG is only approved for infants, because the small available quantity precludes its use for large-scale immunization of adults [13], and equine HBAT, even though protective, is quite reactogenic. Moreover, having a short half-life, it cannot be used prophylactically. Monoclonal antibody (mAb) therapies represent a precise approach to neutralizing BoNTs. They effectively neutralize systemic toxins, thus preventing neuroparalysis in preclinical studies [71]. Two 3-mAb combinations are in development that specifically neutralize BoNT serotypes A and B. The three mAb combinations addressing a single serotype provided successful pre-exposure prophylaxis with one intramuscular injection in the guinea pig inhalation model [72]. Humanized or human mAbs offer advantages over equine-derived antitoxins, such as reduced hypersensitivity risks, unlimited production, and higher specificity. Current studies are exploring cocktails for broader serotype coverage, including A, B, and E. Despite they being very promising, mAb therapies remain in early development.

#### 7.2.4. Camelid Nanobodies

Some camelids, as *Camelus dromedarius* (one hump Arabian camel), llama (*Lama glama* and *L. guanicoe*), and alpaca (*Vicugna pacos*), have the characteristic of having in their serum both a conventional four-chain immunoglobulin and an atypical antibody, called heavy-chain antibody (HCAb), the molecule of which does not contain a light chain [73]. The antigen-binding site of HCAbs is called VHH, which has a sequence homology with the VH of the human antibodies of >80%; thus, it is poorly reactogenic and well tolerated. Moreover, the low molecular weight (approximately 15–20 kDa) makes the single-domain antibody (sdAb) highly penetrable into an active site pocket of a target enzyme, which would never be reached by a large, conventional antibody [73]. Considering that the enzymatically active LC component of BoNT exposes its antigen only after having been translocated into the cytoplasm of the neuronal cell, it becomes inaccessible to conventional high-molecular-weight antibodies [73]. However, the sdAb can be engineered into a “Transbody”, a cell-penetrating antibody, by linking it to a brief sequence of cell penetrating peptide (CPP), which has the capacity of driving its cargo into the cytoplasm, without inducing any cell damage [73].

Recently, six sdAbs, or nanobodies, have been linked on the same heteromultimer, enabling a single heterohexamer protein, delivered via one intramuscular injection, to target six nanobodies against BoNT serotypes A, B, and E. The six nanobodies have also been linked to a replicating RNA and intramuscularly injected in the mice. The treated mice were protected from a challenge with 100 MIPLD50 (mouse intraperitoneal LD_50_) of each BoNT serotype, thus demonstrating the feasibility and the effectiveness of this very promising passive immunization [74].

### 7.3. Small-Molecule Inhibitors

The study on the mechanism of action of BoNTs has allowed the expansion of new strategies for developing more effective inhibitors and antidotes. It has been demonstrated how the host thioredoxin reductase (TxR) is involved in the reduction in the interchain disulfide bridge, highly conserved in all BoNT structures [75]. This enzyme complex is responsible for the biochemical event that leads to the activation of the L in the cytoplasm. Given that the disulfide bridge reduction is a common step for all BoNT serotypes, selective inhibitors of the oxidoreductive system (thioredoxin–thioredoxin reductase, Tx-TxR) could therefore be valid candidates as effective antitoxins.

Small molecules like Auranofin, Curcumin, Myricetin, Juglone, Quercetin, PX-12, and Ebselen were studied as potential inhibitors of this system, also evaluating their toxic power in the consideration of their possible future application in vivo, in humans [76,77,78,79,80,81]. In a precedent study, it was demonstrated in vitro (in intoxicated cerebellar granular neurons) and in vivo (with the complete blockade of the evoked end-plate potential) that all these compounds showed great efficacy in preventing the severity and the duration of the peripheral neuroparalysis caused by the different BoNT serotypes [82], probably because Ebselen acts as a competitive inhibitor of the entire Trx-TrxR system, blocking both subunits of this redox complex, and it has been shown to be the most effective. Moreover, it was demonstrated that a pretreatment with Ebselen 7.5 mg/kg is able to prevent the lethality of the injection of 2 fold MLD_50_ (a double lethal dose for 50% of mice) of BoNTs in adult mice [83]. In the last decade, Ebselen has long been tested and is still under validation for human therapy. During these studies, a substantial record of safety has been reported. Ebselen has been studied in several phase 3 trials in Japan as a neuroprotective treatment to limit the neurological deficits produced by acute stroke and in several phase 2 trials in the USA as a treatment for hearing loss and tinnitus disorders [84,85]. Moreover, the toxicity of Ebselen at various doses and with extended treatments has been thoroughly evaluated in clinical studies investigating its potential use in conditions such as stroke, reperfusion ischemia, acute hearing loss, bipolar disorder, and diabetes. These studies have consistently demonstrated that Ebselen is not toxic in humans, supporting its safety profile for therapeutic use [86,87,88]. This inhibitor showed a role in the prevention of BoNTs’ action only before the LC release; it is not effective once the L chain is already activated in the cytosol. Otherwise, since this inhibitor acts on a common step for all BoNTs, it may be used immediately, without waiting for the serotype identification. Its contribution is more important considering the highly numerous BoNT serotypes (>40 types already reported).

The main field of application of this inhibitor is in the infant botulism treatment, where there is a continuous release of BoNT from the vegetative bacteria implanted into the intestine [89], or in the adult intestinal botulism in immunocompromised patients or in patient who present a weakened intestinal flora, e.g., after recent antimicrobial treatments or gastrointestinal abnormalities. In addition, these compounds could be used as preventive therapy for individuals, as military forces, who have to operate in environments where BoNTs could be released [82]. Table 5 reports the main prophylactic and therapeutic tools.

To confirm the efficacy of Ebselen in preventing and treating botulism in humans, in light of preclinical findings [82,83,86], the initiation of a clinical trial with healthy volunteers is currently under consideration.

## 8. Conclusions

BoNTs are among the most toxic natural substances known and are widely present in the environment, often transmitted through invertebrates, which are resistant to the toxin and may transmit the disease to wild and domesticated animals, which die most frequently during outbreaks. Botulism is a rare but potentially lethal disease in humans, who may be infected/intoxicated through different routes of contamination. Despite the mechanism of action of BoNTs has been known for decades, specific prophylactic and therapeutic tools remain limited. A pentavalent (A–E) vaccine, once used under IND application in selected, limited categories of personnel, has been definitely withdrawn, even for declining potency, and it has not been replaced yet, even though the catalytically inactive recombinant holoprotein candidate vaccine appears to be promising. MoAbs are still in a pre-clinical phase of development. The human antibodies collected from vaccinated individuals are present in limited quantities and may only be used for infant botulism, whereas the heptavalent equine F(ab’)^2^ antibody fragments are protective and even reactogenic. All these preventive and therapeutic tools, being based on a specific immune recognition, are only active against the specific serotype. The small-molecule inhibitors, and among these, in particular Ebselen, a synthetic seleno-organic compound, with electrophilic and potential anti-oxidant activity, is characterized by its safety, ability to inhibit BoNT activation regardless of the serotype, and cost-effective production. It is, therefore, very promising, because it may be administered prophylactically, without the fear of reactogenicity or of inducing an active immunization, which may preclude any future possibility of using BoNT/A or B for therapeutic and/or cosmetic purposes. However, it, like all the other prophylactic or therapeutic tools, may only be active before the BoNT is translocated inside the neuronal cell cytoplasm, while camelid sdAb nanobodies can penetrate the cytoplasm. Despite the BoNTs having been included among the category A biological agents from the CDC, the rarity of botulism classifies it as an orphan disease, lacking effective prophylactic or therapeutic tools.

## Figures and Tables

**Figure 1 biomedicines-13-00411-f001:**
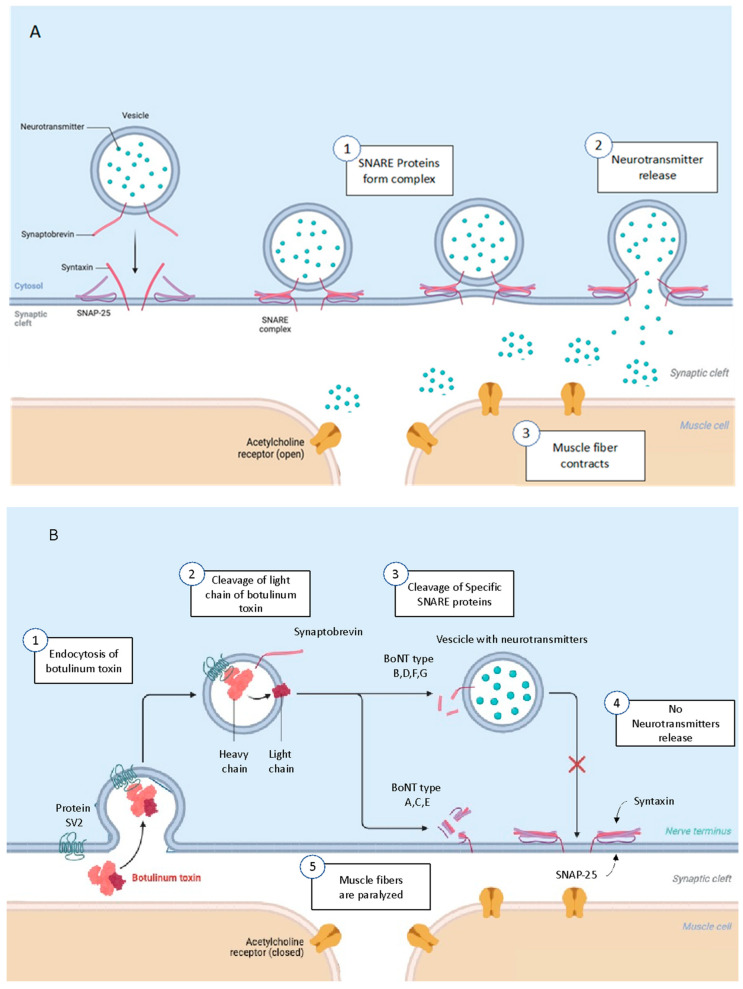
The mechanism of action of the botulinum neurotoxin. (**A**) Normal neurotransmitter release; (**B**) block of release of acetylcholine as a consequence of cleavage of protein complex SNARE by botulinum neurotoxin (created with BioRender.com).

**Table 1 biomedicines-13-00411-t001:** Neurotoxin serotypes, subtypes, and associated botulism (from Ref. [1], slightly modified).

C. botulinum Groups	BoNT Serotypes	Subtypes	Associated Botulism
Group I *C. botulinum* (proteolytic)	A, proteolytic B, F, H	A1, A2, A3, A4, A5, A6, A7, A8, B1, B2, B3, B4, B5(Ba), B6, B7, F1, F2, F3, F4, F5, Ab, Af, Bf, A(B), FA	Human
Group II *C. botulinum* (non-proteolytic)	E, non-proteolytic B, F	B4, E1, E2, E3, E6, E7, E8, E9, E10, E11, F6	Human
Group III *C. botulinum*	C, D	C, D, CD, DC	Animals only (C = birds; D = cattle)
Group IV *C. argentinense* (proteolytic)	G	G	Environmentally isolated only
*C. baratii*	F	F7	Human
*C. butyricum*	E	E4, E5	Human

**Table 2 biomedicines-13-00411-t002:** Summary of botulin neurotoxins as therapeutic and aesthetic tools.

Category	Indications	FDA-Approved Products	Notes
Therapeutic Use	Treatment of overactive bladder (OAB) in adults who have an inadequate response to or are intolerant of an anticholinergic medication Treatment of urinary incontinence due to detrusor overactivity associated with a neurologic condition in adults who have an inadequate response to or are intolerant of an anticholinergic medication Treatment of neurogenic detrusor overactivity (NDO) in pediatric patients 5 years of age and older who have an inadequate response to or are intolerant of anticholinergic medication Prophylaxis of headaches in adult patients with chronic migraine Treatment of spasticity in patients 2 years of age and older Treatment of cervical dystonia in adult patients Treatment of severe axillary hyperhidrosis Treatment of blepharospasm associated with dystonia in patients 12 years of age and older Treatment of strabismus in patients 12 years of age and older	- BoNT/A products: OnabotulinumtoxinA, AbobotulinumtoxinA, and IncobotulinumtoxinA - BoNT/B product: Rimabotulinumtoxin B (only for cervical dystonia)	- Significant therapeutic advancements for conditions with no prior treatment - Acceptable safety, with transient and non-severe adverse events reported - The indications are not exactly the same among the three BoNT/A products.
Esthetic Use	Reduction in facial wrinkles	- BoNT/A products: same as above + DaxibotulinumtoxinA and PrabotulinumtoxinA	- Proven efficacy in wrinkle reduction (4 weeks) - Probable increased risk of ptosis

FDA = Food and Drug Administration; BoNT/A = Botulinum Neurotoxin A; BoNT/B = Botulinum Neurotoxin B.

**Table 3 biomedicines-13-00411-t003:** The characteristics of the different types of botulism.

Forms of Botulism	Source of Contamination	Epidemiology	Incubation Time	Lethality
Foodborne	Ingestion of BoNT-contaminated food	The most frequent but not in the USA	12–72 h	5–10%
Infant	Ingestion of spore-contaminated food in infants up to 1 year of age	The most frequent in the USA	18–36 h	Very low
Adult intestinal	Intestinal colonization in >1-year-old children and adults	Very rare	Unknown	33%
Wound	Wound contamination by spores, mainly in IDU	Increasing frequency among IDU	4–14 days	11%
Iatrogenic	Improper excessive amount of toxin administration	Increasing frequency mainly for cosmetic use	Unknown	Very low
Inhalational	Accidental or intentional as biological weapon	Very rare	72 h	Unknown

BoNT = botulinum neurotoxin; IDU = injecting drug users.

**Table 4 biomedicines-13-00411-t004:** Comparative analysis of different diagnostic methods.

Method	LOD	Turnaround Time	Serotypes Detected	Advantages	Limitations
Mouse bioassay	~0.1 LD_50_/mL	24–48 h	A–G	High sensitivity, gold standard	Ethical concerns, slow
Endopep-MS	0.05–0.1 ng/mL	4–6 h	A, B, E, F	Rapid, serotype-specific	Requires specialized equipment
RT-PCR	~100 copies	2–4 h	A, B, E, F	Gene-specific, fast	Cannot confirm toxin activity
ELISA	10–50 pg/mL	2–4 h	A, B, E, F	Ethical, cost-effective	Antibody cross reactivity
NMJ models	~0.1 ng/mL	4–6 h	A, B	Ethical, high-throughput	Limited validation
Experimental paper-based sensor	0.5–1 nM	1–4 h	A, C	Rapid, cost-effective, smartphone-assisted device, serotype-specific	In validation, a shelf-life of 21 days

LOD = limit of detection; Endopep-MS = Endopeptidase Mass Spectrometry; RT-PCR = Reverse Transcriptase-Polymerase Chain Reaction; ELISA = Enzyme-Linked Immunosorbent Assay; NMJ = neuromuscular junction.

**Table 5 biomedicines-13-00411-t005:** BoNT prophylactic and therapeutic tools not available anymore, currently available, or in development.

Therapy	Mechanism	FDA Approval	Target Serotypes	Patient Effects	Limitations
Pentavalent Vaccine	Active immunization	No	A–E	Provides long-term immunity for high-risk individuals	Indicated for exposed personnel (laboratory workers and the military)
BIG-IV (BabyBIG)	Neutralizes circulating toxins	Yes	A, B	Shortens hospital stays, reduces ventilator dependency, prevents paralysis	Restricted to infants; no effect on internalized toxins
HBAT	Neutralizes circulating toxins	Yes	A–G	Mitigates systemic progression, reduces respiratory failure risk	Reactogenicity (risk of hypersensitivity and serum sickness)
Monoclonal Abs	Targets BoNT specific epitopes	No	A, B, E (potentially all the serotypes)	Neutralizes circulating toxins, improves survival in animal models	Developmental preclinical stage
Ebselen	Irreversibly inhibits protease	No	Not serotype-specific	Reduces neuroparalysis, delays paralysis onset	Preclinical stage

BIG-IV = Human Botulism Immune Globulin Intravenous; HBAT = Heptavalent Botulism Anti-Toxin.

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
