# Peer review of "Botulinum Neurotoxins as Two-Faced Janus Proteins"

_biomedicines, 2025, doi:10.3390/biomedicines13020411_

Round 1
Reviewer 1 Report
Comments and Suggestions for Authors
The manuscript is well written and the subject is highly relevant. I will only make a few minor suggestions.
L.54-55: Has this new botulinum-like neurotoxin been associated with any cases of intoxication, or has only the chemical characterization of the toxin been performed?
L.244: It would be interesting to present the main characteristics of each product due to the differences in manufacturing.
L.336: This sentence should be placed after describing all types of botulism.
L.339: I believe it is better to call it Accidental inhalation botulism to differentiate it from cases of use as a biological weapon.
L.343-428: The epidemiology section could be placed after the biological weapon item. Intentional use of the toxin as a weapon causes a type of botulism, and I believe it could be included in Table 3. Or item 5 could be renamed accidental or unintentional botulism.
Author Response
We wish to thank the reviewer for dedicating time to reviewing our study and for the valuable comments and suggestions. We have carefully considered and incorporated the requested changes, enhancing the clarity of our work.
In the revised manuscript, the text that has been added or modified is written in red.
Comment 1: The manuscript is well written and the subject is highly relevant. I will only make a few minor suggestions. L.54-55: Has this new botulinum-like neurotoxin been associated with any cases of intoxication, or has only the chemical characterization of the toxin been performed?
Response 1: No, to date, there have been no reported cases of botulism in either animals or humans associated with the new botulinum-like neurotoxins.
Comment 2: L.244: It would be interesting to present the main characteristics of each product due to the differences in manufacturing.
Response 2: This information has been added to paragraph 4, lines 260–271 of the revised text:
"The three main BoNT/A therapeutic products (OnabotulinumtoxinA, AbobotulinumtoxinA, and IncobotulinumtoxinA) currently available are not interchangeable due to differences in manufacturing [29]. These differences primarily concern the purification method—crystallization for OnabotulinumtoxinA and chromatography for the other two products—the composition of the final product (OnabotulinumtoxinA and AbobotulinumtoxinA contain inactive complex proteins, with a final molecular weight of ~900 kDa and <500 kDa, respectively, whereas IncobotulinumtoxinA consists solely of the ~150 kDa toxin), the potency of the three products (determined using manufacturer-specific assays), and the dosage units. OnabotulinumtoxinA and IncobotulinumtoxinA are available in 50, 100, and 200-unit formulations, while AbobotulinumtoxinA is available in 300- and 500-unit formulations [29]. However, even for products with identical dosage units, they are not interchangeable, as indicated in the drug labels, due to differences in potency—these are biological products rather than chemically identical compounds [29]."
Furthermore, to ensure a smooth transition between topics without interrupting the newly added explanation, the sentence originally written in lines 244–245 has been modified and moved to lines 272–274 of the revised text as follows:
"In addition to these formulations, BoNT/B has also been approved for therapeutic use. Specifically, RimabotulinumtoxinB has been approved for the treatment of cervical dystonia [30]."
Comment 3: L.336: This sentence should be placed after describing all types of botulism.
Response 3: This sentence has been placed after the description of all types of botulism, at line 459 of the revised text. Additionally, Table 3 has been moved to the same location.
Comment 4: L.339: I believe it is better to call it Accidental inhalation botulism to differentiate it from cases of use as a biological weapon.
Response 4: Item 5.6 has been renamed according to the route of exposure, reflecting that inhalation botulism may occur either accidentally or intentionally, as stated in the text and in Table 3.
Comment 5: L.343-428: The epidemiology section could be placed after the biological weapon item. Intentional use of the toxin as a weapon causes a type of botulism, and I believe it could be included in Table 3. Or item 5 could be renamed accidental or unintentional botulism.
Response 5: According to the reviewer’s suggestion, the biological weapon section has now been placed before the epidemiological section, as section 5.7.
Consequently, the epidemiology paragraph is now section 5.8, and all subsequent sections have been renumbered accordingly.
Reviewer 2 Report
Comments and Suggestions for Authors
The article focuses on botulinum neurotoxins (BoNT), discussing their toxicity, mechanisms, and applications. It highlights how BoNT induces muscle paralysis by blocking acetylcholine release and its extensive use in treating spastic disorders and in the cosmetic field. Despite its significant medical and cosmetic value, there are risks of iatrogenic botulism caused by counterfeit products. The article provides a detailed overview of BoNT's structure, genetic characteristics, diagnostic methods, and therapeutic approaches, such as FDA-approved BIG-IV and HBAT, as well as emerging therapies like monoclonal antibodies, nanobodies, and Ebselen. Additionally, it explores the history and potential threat of BoNT as a biological weapon, emphasizing its importance in public health and biosecurity.
The article covers a broad scope, but some sections suffer from redundancy. For instance, the mechanism section elaborates extensively on SNARE protein function. Additionally, it briefly mentions wound infections caused by spore contamination but lacks in-depth discussion on the conditions necessary for toxin production (e.g., oxygen levels and tissue environment). There is a wealth of literature on wound-related signaling pathways, such as TNFα/AKT-β-catenin (Wang et al., 2017; Han et al., 2023), which could have enriched the discussion. Recent advances in skin inflammation studies should also be considered. During inflammatory responses like wounds, macrophage numbers increase, particularly CX3CR1 bone-marrow-derived macrophages (Wang, X., Nat Commun 2017), releasing TNF and TGFβ1 (Rahmani, W., J Invest Dermatol 2018). TNF, via the AKT/β-catenin signaling axis, stimulates hair follicle stem cells (HFSCs), inducing wound-induced hair anagen re-entry/growth (WIHA) and wound-induced hair follicle neogenesis (WIHN). The TGFβ1 signal is critical in WIHA/WIHN, potentially initiating hair follicle regeneration through the AKT/PI3K pathway (Chen, H., PLoS One 2017; Chen, H., Theranostics 2019). Furthermore, some references in the article appear outdated and fail to include findings from 2023.
Author Response
We wish to thank the reviewer for dedicating time to reviewing our study and for the valuable comments and suggestions. We have carefully considered the requested changes and have modified the manuscript to improve the clarity of our work while maintaining the balance of the sections.
In the revised manuscript, the text that has been added or modified is written in red.
Comment 1: The article focuses on botulinum neurotoxins (BoNT), discussing their toxicity, mechanisms, and applications. It highlights how BoNT induces muscle paralysis by blocking acetylcholine release and its extensive use in treating spastic disorders and in the cosmetic field. Despite its significant medical and cosmetic value, there are risks of iatrogenic botulism caused by counterfeit products. The article provides a detailed overview of BoNT's structure, genetic characteristics, diagnostic methods, and therapeutic approaches, such as FDA-approved BIG-IV and HBAT, as well as emerging therapies like monoclonal antibodies, nanobodies, and Ebselen. Additionally, it explores the history and potential threat of BoNT as a biological weapon, emphasizing its importance in public health and biosecurity. The article covers a broad scope, but some sections suffer from redundancy. For instance, the mechanism section elaborates extensively on SNARE protein function.
Response 1: We thank the reviewer for pointing out a redundancy that we had not noticed. Specifically, the following sentence: “SNARE proteins allow the opening of membrane vesicles containing acetylcholine and the fusion with the membrane of the pre-synaptic neuronal cell, thus releasing acetylcholine at the level of the synaptic cleft, thereby inducing muscle activation and contraction,” originally found in lines 144–146, has been removed, as it is a complete repetition of the content already stated in lines 134–137.
Comment 2: Additionally, it briefly mentions wound infections caused by spore contamination but lacks in-depth discussion on the conditions necessary for toxin production (e.g., oxygen levels and tissue environment). There is a wealth of literature on wound-related signaling pathways, such as TNFα/AKT-β-catenin (Wang et al., 2017; Han et al., 2023), which could have enriched the discussion. Recent advances in skin inflammation studies should also be considered. During inflammatory responses like wounds, macrophage numbers increase, particularly CX3CR1 bone-marrow-derived macrophages (Wang, X., Nat Commun 2017), releasing TNF and TGFβ1 (Rahmani, W., J Invest Dermatol 2018). TNF, via the AKT/β-catenin signaling axis, stimulates hair follicle stem cells (HFSCs), inducing wound-induced hair anagen re-entry/growth (WIHA) and wound-induced hair follicle neogenesis (WIHN). The TGFβ1 signal is critical in WIHA/WIHN, potentially initiating hair follicle regeneration through the AKT/PI3K pathway (Chen, H., PLoS One 2017; Chen, H., Theranostics 2019). Furthermore, some references in the article appear outdated and fail to include findings from 2023.
Response 2: The following sentence has been added to the paragraph on wound botulism, at lines 340–342 of the revised manuscript:
"The conditions necessary for toxin production in wound botulism are strict anaerobiosis, pH >4.6, and high protein concentration [Wong DM, Young-Perkins KE, Merson RL. Factors influencing Clostridium botulinum spore germination, outgrowth, and toxin formation in acidified media. Appl Environ Microbiol. 1988 Jun;54(6):1446-50. doi: 10.1128/aem.54.6.1446-1450.1988], all of which are found in the wound environment."
Moreover, additional specifications regarding germination have been included at lines 208–215 of the revised manuscript.
Regarding the suggestion to include a discussion on macrophage activation and the biochemical signaling pathways triggered by the inflammatory response in the wound, along with stem cell activation driving wound healing, we encountered some difficulties in incorporating this aspect into the text. While the referenced papers are of high quality, they do not appear to be directly related to the topic of our paper.
Although macrophage activation could hypothetically influence wound botulism by either reducing wound healing time—thereby theoretically lowering the risk of wound contamination—or by providing a defense mechanism against C. botulinum alongside initiating the wound healing process, these possibilities seem unlikely based on the following considerations:
- The highest current risk for wound botulism is among injectable drug users, particularly those using black tar heroin via skin-popping. In this context, there is a high probability that the heroin becomes contaminated during dilution, after manufacturing, or during distribution, but in any case, before injection [Passaro DJ, Werner SB, McGee J, Mac Kenzie WR, Vugia DJ. Wound botulism associated with black tar heroin among injecting drug users. JAMA. 1998 Mar 18;279(11):859-63. doi: 10.1001/jama.279.11.859].
- Regarding the potential macrophage activation-mediated defense against Clostridium species, there is no direct evidence in the literature, apart from the general role of macrophages in first-line anti-infectious defense [Park MD, Silvin A, Ginhoux F, Merad M. Macrophages in health and disease. Cell. 2022 Nov 10;185(23):4259-4279. doi: 10.1016/j.cell.2022.10.007].
As for the observations related to findings from 2023, if they pertain to the same topic, the above considerations remain applicable.
Round 2
Reviewer 2 Report
Comments and Suggestions for Authors
According to all the reviewer's opinions, I think this paper is ready for publication